# MASKS, SIGNS, AND LEARNING RATE REWINDING

**Advait Gadhikar & Rebekka Burkholz**
CISPA Helmholtz Center for Information Security
Saarbrücken, Germany
{advait.gadhikar, burkholz}@cispa.de

## ABSTRACT

Learning Rate Rewinding (LRR) has been established as a strong variant of Iterative Magnitude Pruning (IMP) to find lottery tickets in deep overparameterized neural networks. While both iterative pruning schemes couple structure and parameter learning, understanding how LRR excels in both aspects can bring us closer to the design of more flexible deep learning algorithms that can optimize diverse sets of sparse architectures. To this end, we conduct experiments that disentangle the effect of mask learning and parameter optimization and how both benefit from overparameterization. The ability of LRR to flip parameter signs early and stay robust to sign perturbations seems to make it not only more effective in mask identification but also in optimizing diverse sets of masks, including random ones. In support of this hypothesis, we prove in a simplified single hidden neuron setting that LRR succeeds in more cases than IMP, as it can escape initially problematic sign configurations.

## 1 INTRODUCTION

Overparametrization has been key to the huge success of deep learning (Bubeck et al., 2023; Neyshabur et al., 2019; Belkin et al., 2019). Adding more trainable parameters to models has shown to consistently improve performance of deep neural networks over multiple tasks. While it has been shown that there often exist sparser neural network representations that can achieve competitive performance, they are usually not well trainable by standard neural network optimization approaches (Evci et al., 2022), which is a major challenge for learning small scale (sparse) neural networks from scratch to save computational resources.

The Lottery Ticket Hypothesis (LTH) by Frankle & Carbin (2019) is based on an empirical existence proof that the optimization of at least some sparse neural network architectures is feasible with the right initialization. According to the LTH, dense, randomly initialized neural networks contain subnetworks that can be trained in isolation with the same training algorithm that is successful for the dense networks. A strong version of this hypothesis Ramanujan et al. (2020a); Zhou et al. (2019), which has also been proven theoretically (Malach et al., 2020; Pensia et al., 2020; Orseau et al., 2020; Fischer et al., 2021; Burkholz et al.; Burkolz, 2022; da Cunha et al., 2022; Gadhikar et al., 2023; Ferbach et al., 2023), suggests that the identified initial parameters might be strongly tied to the identified sparse structure. Related experimental studies and theoretical investigations support this conjecture (Evci et al., 2022; Paul et al., 2023).

In line with these findings, contemporary pruning algorithms currently address the dual challenge of structure and parameter learning only jointly. Iterative Magnitude Pruning (IMP) (Frankle & Carbin, 2019) and successive methods derived from it, like Weight Rewinding (WR) (Frankle et al., 2020a) and Learning Rate Rewinding (LRR) (Renda et al., 2020; Liu et al., 2021a) follow an iterative *pruning – training* procedure that removes a fraction of parameters in every pruning iteration until a target sparsity is reached. This achieves state-of-the-art neural network sparsification (Paul et al., 2023), albeit at substantial computational cost.

While this cost can be reduced by starting the pruning procedure from a sparser, randomly pruned network (Gadhikar et al., 2023), the question remains whether the identification of small sparse neural network models necessitates training an overparameterized model first. Multiple works attest that overparameterization aids pruning (Zhang et al., 2021; Chang et al., 2021; Golubeva et al., 2020).

This suggests that overparameterized optimization obtains information that should be valuable for the performance of a sparsified model. Conforming with this reasoning, IMP was found less effective for complex architectures than Weight Rewinding (WR) (Renda et al., 2020), which rewinds parameters to values that have been obtained by training the dense, overparameterized model for a few epochs (instead of rewinding to their initial value like IMP). LRR (Renda et al., 2020) completely gets rid of the weight rewinding step and continues to train a pruned model from its current state while repeating the same learning rate schedule in every iteration. Eliminating the parameter rewinding step has enabled LRR to achieve consistent accuracy gains and improve the movement of parameters away from their initial values (Liu et al., 2021a).

Complimentary to Paul et al. (2023); Liu et al. (2021a), we identify a mechanism that provides LRR with (provable) optimization advantages that are facilitated by pruning a trained overparameterized model. First, we gain provable insights into LRR and IMP for a minimal example, i.e., learning a single hidden ReLU neuron. Our exact solutions to the gradient flow dynamics for high-dimensional inputs could be of independent interest. The initial overparameterization of the hidden neuron enables learning provably and facilitates the identification of the correct ground truth mask by pruning. LRR benefits from the robustness of the overparameterized neuron to different parameter initializations, as it is capable of switching initially problematic parameter sign configurations that would result in the failure of IMP.

We verify in extensive experiments on standard benchmark data that our theoretical insights capture a practically relevant phenomenon and that our intuition regarding parameter sign switches also applies to more complex architectures and tasks. We find that while LRR is able to perform more sign flips, these happen in early training – pruning iterations, when a higher degree of overparameterization is available to facilitate them. In this regime, LRR is also more robust to sign perturbations.

This observation suggests that LRR could define a more reliable parameter training algorithm than IMP for general masks. However in iterative pruning schemes like IMP and LRR, the mask identification step is closely coupled with parameter optimization. Changing either of these aspects could affect the overall performance considerably. For example, learning only the mask (strong lottery tickets (Ramanujan et al., 2020b)) or learning only the parameters with a random mask (Liu et al., 2021b; Gadhikar et al., 2023) are unable to achieve the same performance as IMP at high sparsities. Yet, we carefully disentangle the optimization of parameters and mask learning aspect to show that LRR achieves more reliable training results for different masks. In addition, it can also identify a better mask that can sometimes achieve a higher performance than the IMP mask, even when both are optimized even with IMP.

**Contributions.** Our main contributions are as follows:

- To analyze the advantages of LRR for parameter optimization and mask identification, we conduct experiments that disentangle these two aspects and find that the benefits of LRR are two-fold. (a) LRR often finds a better sparse mask during training and (b) LRR is more effective in optimizing parameters of a diverse masks (eg: a random mask).

- We experimentally verify that, in comparison with IMP, LRR is more flexible in switching parameter signs during early pruning iterations, when the network is still overparameterized. It also recovers more reliably from sign perturbations.

- For a univariate single hidden neuron network, we derive closed form solutions of its gradient flow dynamics and compare them with training and pruning an overparameterized neuron. LRR is provably more likely to converge to a ground truth target while IMP is more susceptible to failure due to its inability to switch initial problematic weight signs.

## 1.1 RELATED WORK

**Insights into IMP.** Paul et al. (2023) attribute the success of IMP to iteratively pruning a small fraction of parameters in every step which allows consecutively pruned networks to be linearly mode connected (Frankle et al., 2020a; Paul et al., 2022). This can be achieved by WR if the dense network is trained for sufficiently many epochs. They argue that as long as consecutive networks are sufficiently close, IMP finds sparse networks that belong to the same linearly mode connected region of the loss landscape. Evci et al. (2022) similarly claim that IMP finds an initialization that is close to the pruning solution and within the same basin of attraction. Liu et al. (2021a) similarly show

that initial and final weights are correlated for IMP. In our experiments we study the WR variant of IMP, where the dense network has been trained for sufficiently many epochs to obtain the initial parameters for IMP, but we still find that, in comparison, LRR switches more signs and can achieve better performance.

**The role of sign switches.** While Wang et al. (2023) have recently verified the importance of suitable parameter signs for better training of neural networks in general, they have not analyzed their impact on neural network sparsification. Zhou et al. (2019) study the weight distributions for IMP and find that rewinding only parameter signs can be sufficient. Large scale problems, however, rely on learning signs in early epochs and require a good combination with respective parameter magnitudes, as discussed by Frankle et al. (2020b) for IMP. These results are still focused on the IMP learning mechanism and its coupling to the mask learning. In contrast, we show that identifying good signs (and magnitudes) early enables LRR to not only find a better mask but to also learn more effectively if the mask identification is independent from the parameter optimization.

**Mask optimization.** Random sparse masks also qualify as trainable lottery tickets Su et al. (2020); Ma et al. (2021); Liu et al. (2021b) which suggests that the mask identification can be separated from parameter optimization upto certain sparsities (Gadhikar et al., 2023). Our experiments isolate the advantages of LRR on both these aspects.

**Training dynamics of overparametrized networks.** The training dynamics of overparametrized networks have been theoretically investigated in multiple works, which frequently employ a balanced initialization (Du et al., 2018) and a related conservation law under gradient flow in their analysis. Arora et al. (2018; 2019) study deep linear networks in this context, while Du et al. theoretically characterizes the gradient flow dynamics of two layer ReLU networks. While they require a high degree of overparameterization, Boursier et al. (2022) obtains more detailed statements on the dynamics with a more flexible paramterization but assume orthogonal data input.

**Single hidden neuron setting.** These results do not directly transfer to the single hidden neuron case, which has been subject of active research Yehudai & Ohad (2020); Lee et al. (2022a); Vardi et al. (2021); Oymak & Soltanolkotabi (2019); Soltanolkotabi (2017); Kalan et al. (2019); Frei et al. (2020); Diakonikolas et al. (2020); Tan & Vershynin (2019); Du et al.. Most works assume that the outer weight $a$ is fixed, while only the inner weight vector $\boldsymbol{w}$ is learned and mostly study noise free data. We extend similar results to trainable outer weight and characterize the precise training dynamics of an univariate (masked) neuron in closed form. Lee et al. (2022b) study a similar univariate case but do not consider label noise in their analysis.

Most importantly, similar results have not been deduced and studied under the premise of network pruning. They enable us to derive a mechanism that gives LRR a provable benefit over IMP, which is inherited from overparameterized training.

## 2 THEORETICAL INSIGHTS FOR A SINGLE HIDDEN NEURON NETWORK

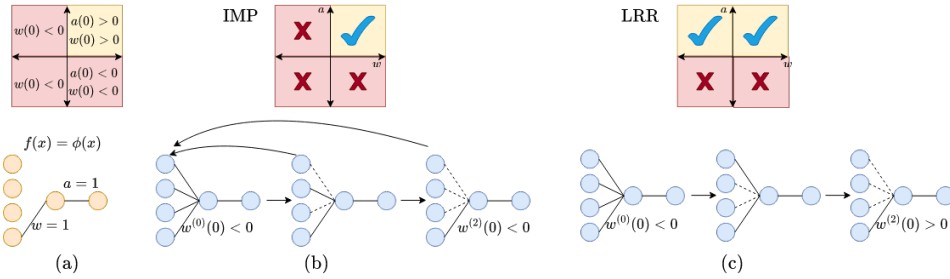

Figure 1: (a) Target network. For one dimensional input, learning succeeds when the initial values $w(0), a(0) > 0$ are both positive (yellow quadrant), but fails in all other cases (red). (b) For multidimensional input, IMP identifies the correct mask, but cannot learn the target if the model is reinitialized to $w^{(2)}(0) < 0$. (c) LRR identifies the correct mask and is able to inherit the correct initial sign $w^{(2)}(0) > 0$ from the trained overparameterized model if $a^{(0)}(0) > 0$.

**Intuition behind LRR versus IMP.** The advantage of IMP is that it was designed to identify lottery tickets and thus successfully initialize sparse masks (i.e. sparse neural network structures). However, in order to find such an initialization, we show that the information obtained in earlier pruning iterations with the aid of overparameterization is valuable in learning better models. Notably, we find that each pruning iteration transfers key information about parameter signs to the next iteration. Forgetting this information (due to weight rewinding) means that IMP is challenged to learn the appropriate parameter signs from scratch in each iteration.

To establish provable insights of this form, we face the theoretical challenge to describe the learning dynamics of the parameters in response to different initializations. We therefore focus on an example of minimum complexity that still enables us to isolate a mechanism by which LRR has a higher chance to succeed in solving a learning task. In doing so, we study a single hidden neuron, 2-layer neural network under gradient flow dynamics, as visualized in Fig. 1 (a).

For our purpose, we focus on two main aspects: (i) The trainability of the masked neural network (i.e., a single hidden neuron with $d = 1$ input), once the sparse mask is identified. (ii) The ability of LRR to leverage the initial overparameterization (i.e., a single hidden neuron with $d > 1$ inputs) in the model to learn appropriate parameter signs.

Regarding (i), we have to distinguish four different initialization scenarios. Only one scenario (yellow quadrant in Fig. 1 (a)) leads to accurate learning. LRR is able to inherit this setup from the trained overparameterized network and succeed (see Fig. 1 (c)) in a case when IMP fails (Fig. 1 (b)) because it rewinds its parameters to an initial problematic setting. To explain these results in detail, we have to formalize the set-up.

**LRR.** We focus on comparing Iterative Magnitude Pruning (IMP) and Learning Rate Rewinding (LRR). Both cases comprise iterative pruning-training cycles. The $i$-th pruning cycle identifies a binary mask $M^{(i)} \in \{0,1\}^N$, which is established by pruning a fraction of neural network parameters $\theta^{(\mathbf{i-1})}$ with the smallest magnitude. A training cycle relearns the remaining parameters $\theta^{(\mathbf{i})}$ of the masked neural network $f(x \mid \mathbf{M}^{(\mathbf{i})}\theta^{(\mathbf{i})})$. The only difference between LRR and IMP is induced by how each training cycle is initialized (See Fig. 1(b)). In case of LRR, the parameters of the previous training cycle that were not pruned away are used as initial values of the new training cycle so that $\theta^{(i)}(0) = \theta^{(i-1)}(t_{end})$. Thus, training continues (with a learning rate that is reset to its initial value).

**IMP.** In case of IMP, each pruning iteration starts from the same initial parameters $\theta^{(i)}(0) = \theta^{(0)}(0)$ and parameters learnt in the previous iteration are forgotten. While our theoretical analysis focuses on IMP, Frankle et al. (2021); Su et al. (2020); Ma et al. (2021) have shown that IMP does not scale well to larger architectures. Hence, we employ the more successful variant Weight Rewinding (WR) in our experiments. Here, the parameters are not rewound to their initial values but to the parameters of the dense network which was trained for a few warm-up epochs $\theta^{(i)}(0) = \theta^{(0)}(k)$ Frankle et al. (2020a); Renda et al. (2020). Our theory also applies to this case but we will mostly discuss rewinding to the initial values for simplicity. From now on we use IMP to refer to IMP in our theory and WR in our experiments.

**Problem set-up.** Consider a single hidden neuron network with input $\boldsymbol{x} \in \mathbb{R}^d$, given as $f(\boldsymbol{x}) := a\phi(\boldsymbol{wx})$ with the ReLU activation $\phi(x) = \max\{x, 0\}$ (see Fig. 1). Note that one of the weights could assume the role of a bias if one of the inputs is constant in all samples, e.g., $x_i = 1$. The task is to learn a scalar target $t(\mathbf{x}) = \phi(\mathbf{x_1})$ only dependent on the first coordinate of $\boldsymbol{x}$, from which $n$ noisy training data points $Y = t(X_1) + \zeta$ are generated (upper case denotes random variables.) For simplicity, we assume that all input components are independently and identically (iid) distributed and follow a normal distribution $X_i \sim \mathcal{N}(0, \mathbf{I}/d)$, while the noise follows an independent normal distribution $\zeta \sim \mathcal{N}(0, \sigma^2)$. The precise assumptions on the data distributions are not crucial for our results but clarify our later experimental setting. Based on a training set $(\boldsymbol{x}_i, y_i)$ for $i \in [n] = \{1, 2.., n\}$, learning implies minimizing the mean squared error under gradient flow

$$\mathcal{L} = \frac{1}{2n}\sum_{i=1}^{n}\left(f(\boldsymbol{x}_i) - y_i\right)^2, \quad \frac{d\mathcal{L}}{dt} = -\frac{\partial\mathcal{L}}{\partial a}; \quad \frac{dw_i}{dt} = -\frac{\partial\mathcal{L}}{\partial w_i} \quad (\forall i \in [1, d]), \tag{1}$$

which resembles the dynamics induced by minimizing $\mathcal{L}$ with gradient descent for sufficiently small learning rates. Note that also higher learning rates and more advanced optimizers like LBFGS converge to the same values that we derive based on gradient flow for this exemplary problem. Stochastic Gradient (SGD) would introduce additional batch noise and exaggerate the issue that we

will discuss for small sample sizes. As gradient flow is sufficient to highlight the mechanism that we are interested in, we focus our analysis on this case.

To simplify our exposition and to establish closed form solutions, we assume that the parameters are initialized in a balanced state such that $a(0)^2 = \sum_{i=1}^{d} w_i^2(0)$, which is preserved through training (Arora et al., 2018; Du et al.; 2018) so that $a(t)^2 = \sum_{i=1}^{d} w_i^2(t)$.

## 2.1 Training dynamics for one-dimensional input ($d = 1$)

Let us start with the case, in which we have identified the correct mask by pruning away the remaining inputs and we know the ground truth structure of the problem. Studying this one dimensional case will help us identify typical failure conditions in the learning dynamics and how these failure conditions are more likely to occur in IMP than LRR. Knowing the correct mask, our model is reduced to the one-dimensional input case ($d = 1$) after pruning, so that $f(x) = a\phi(wx)$, while the target labels are drawn from $y \sim \phi(x) + \zeta$.

Since the ReLU neuron is active only when $wx > 0$, we have to distinguish all possible initial sign combinations of $w$ and $a$ to analyze the learning dynamics. The following theorem states our main result, which is also visualized in Fig. 1 (a).

**Theorem 2.1.** *Let a target $t(x) = \phi(x)$ and network $f(x) = a\phi(wx)$ be given such that $a$ and $w$ follow the gradient flow dynamics (1) with a random balanced parameter initialization and sufficiently many samples. If $a(0) > 0$ and $w(0) > 0$, $f(x)$ can learn the correct target. In all other cases $(a(0) > 0, w(0) < 0)$, $(a(0) < 0, w(0) > 0)$ and $(a(0) < 0, w(0) < 0)$ learning fails.*

The proof in Appendix A.1 derives the closed form solutions of the learning dynamics of $f(x)$ under gradient flow for each combination of initial signs. It establishes that training a single neuron $f(x) = a\phi(wx)$ from scratch to learn the noisy target $\phi(x) + \zeta$ can be expected to fail at least with probability $3/4$ if we choose a standard balanced parameter initialization scheme where either signs are equally likely to occur for $a(0), w(0)$.

Why should this imply a disadvantage for IMP over LRR? As we will argue next, overparameterization in form of additional independent input dimensions $x \in \mathbb{R}^d$ can improve substantially the learning success as the set of samples activated by ReLU becomes less dependent on the initialization of the first element $w_1(0)$ of $w$. Thus training an overparameterized neuron first, enables LRR and IMP to identify the correct mask. Yet, after reinitialization, IMP is reduced to the failure case described above with probability $3/4$, considering the combination of initial signs of $a(0)$ and $w_1(0)$. In contrast, LRR continues training from the learned parameters. It thus inherits a potential sign switch from $w_1(0) < 0$ to $w_1(0) > 0$ if $a(0) > 0$ during training (and pruning) the overparameterized model. Thus, the probability that LRR fails due to a bad initial sign after identifying the correct mask is reduced to $1/2$, as also explained in Fig. 1.

## 2.2 Learning an overparametrized neuron ($d > 1$)

As we have established the failure cases of the single input case in the previous section, we now focus on how overparameterization (to $d > 1$) can help avoid one case and thus aid LRR, while IMP is unable to benefit from the same.

Multiple works have derived that convergence of the overparameterized model ($d > 1$) happens under mild assumptions and with high probability in case of zero noise and Gaussian input data, suggesting that overparameterization critically aids our original learning problem. For instance, (Yehudai & Ohad, 2020) have shown that convergence to a target vector $v$ is exponentially fast $\|w(t) - v\| \le \|w(0) - v\| \exp(-\lambda t)$, where the convergence rate $\lambda > 0$ depends on the angle between $w(0)$ and $w(t)$ assuming that $a(0) = a(t) = 1$ is not trainable.

**Insight:** For our purpose, it is sufficient that the learning dynamics can change the sign of $w_1(0) < 0$ to $w_1(\infty) > 0$ if $d \ge 2$. This would correspond to the first training round of LRR and IMP. Furthermore, training the neuron with multiple inputs enables the pruning step to identify the correct ground truth mask under zero noise, as $w_k(\infty) \approx 0$ for $k \ne 1$. Yet, while IMP would restart training from $w_1(0) < 0$ and fail to learn a parameterization that corresponds to the ground truth, LRR succeeds, as it starts from $w_1(\infty) > 0$.

These results assume, however, that $a(0) = 1$ is fixed and not trainable. In the previous section, we have also identified major training failure points if $a(0) < 0$. As it turns out, training a single multivariate neuron does not enable recovery from such a problematic initialization in general.

**Lemma 2.2.** *Assume that $a$ and $\boldsymbol{w}$ follow the gradient flow dynamics induced by Eq. (7) with Gaussian iid input data, zero noise, and that initially $0 < |a|\|\boldsymbol{w}(0)\| \le 2$ and $a(0)^2 = \|\boldsymbol{w}(0)\|^2$. Then $a$ cannot switch its sign during gradient flow.*

This excludes another relevant event that could have given IMP an advantage over LRR. Note that IMP could succeed while LRR fails, if we start from a promising initialization $w_1(0) > 0$ and $a(0) > 0$ but the parameters converge during the first training round to values $w_1(0) < 0$ and $a(0) < 0$ that would hamper successful training after pruning. This option is prevented, however, by the fact that $a$ cannot switch its sign in case of zero noise. We therefore conclude our theoretical analysis with our main insight.

**Theorem 2.3.** *Assume that $a$ and $\boldsymbol{w}$ follow the gradient flow dynamics induced by Eq. (7) with Gaussian iid input data, zero noise, and that initially $0 < |a|\|\boldsymbol{w}(0)\| \le 2$ and $a(0)^2 = \|\boldsymbol{w}(0)\|^2$. If $w_1(0) < 0$ and $a(0) > 0$, LRR attains a lower objective (1) than IMP. In all other cases, LRR performs at least as well as IMP.*

### 2.3 VERIFYING THEORETICAL INSIGHTS BASED ON SINGLE HIDDEN NEURON NETWORK

Figure 2 (a) empirically validates our theoretical insights for $d > 1$ and compares LRR and IMP for each combination of initial signs of $a(0), w_1(0)$. A single hidden neuron network with input dimension $d = 10$ and random balanced Gaussian initialization is trained with LBFGS to minimize the objective function 1 for a noisy target ($\sigma^2 = 0.01$). Averages and 0.95 confidence intervals over 10 runs for each case are shown. In each run, we prune and train over 3 levels for 1000 epochs each, while removing the same fraction of parameters in each level to achieve a target sparsity of $90\%$ so that only one single input remains. In line with the theory, we find that IMP is only successful in the case $a(0) > 0$ and $w_1(0) > 0$, while LRR succeeds as long as $a(0) > 0$.

## 3 EXPERIMENTS

Our first objective is to analyze whether our theoretical intuition that LRR is more flexible in learning advantageous sign configurations transfers to more complex tasks related to standard benchmarks. Different from the simplified one hidden neuron setting, LRR and IMP also identify different masks. Thus, our second objective is to disentangle the impact of the different learning mechanisms and potential sign flips on both, the mask learning and the parameter optimization given a fixed mask.

To this end, we perform experiments on CIFAR10, CIFAR100 (Krizhevsky, 2009) and Tiny ImageNet (Le & Yang, 2015) with ResNet18 or ResNet50 with IMP and LRR that start from the same initializations. Table 1 in the appendix describes the details of the setup. To strengthen the IMP baseline, we in fact study WR and thus rewind the parameters to values that we have obtained after a sufficiently high number of training epochs of the dense model, which is in line with successfully obtaining matching networks as found by Paul et al. (2023).

**LRR modifications.** Different from our theoretical investigations, we have to take more complex factors into account that influence the training process like learning rate schedules and batch normalization (BN). We found that the originally proposed training schedule of LRR can suffer from diminishing BN weights that impair training stability on larger scale problems like CIFAR100 and Tiny ImageNet (see Fig. 4 and Fig. 11 in the appendix). To avoid this issue, we propose to rewind BN parameters when the mask is decoupled from parameter optimization. In all our experiments, we introduce warmup after each pruning iteration, which increases the flexibility of LRR to optimize different masks as well as improves baseline performance (see Fig. 8 in appendix). Fig. 4 (c, d) provides an example where these modifications make LRR competitive with IMP on the IMP mask.

We start our investigations with observations regarding the performance of LRR and IMP in different learning scenarios before we isolate potential mechanisms that govern these observations like sign flips and network overparameterization. Our experiments establish and confirm that LRR outperforms IMP on all our benchmarks. Does this performance boost result from an improved mask identification or stronger parameter optimization?

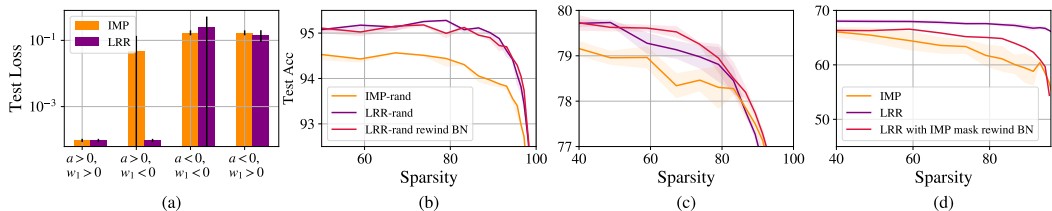

Figure 2: (a) IMP and LRR for a single hidden neuron network. (b, c) Mask randomization for (b) CIFAR10 and (c) CIFAR100. (d) LRR optimizes the IMP mask more effectively on Tiny ImageNet.

**LRR identifies a better mask.** Even though the mask identification of IMP is coupled to its training procedure, Fig. 3 (a, b) show that the mask that has been identified by LRR also achieves a higher performance than the IMP mask on CIFAR10 when its parameters are optimized with IMP. Similar improvements are observed on CIFAR100 (Fig. 3 (c, d)) except at high sparsities ($> 95\%$) where the coupling of the mask and parameter optimization is more relevant.

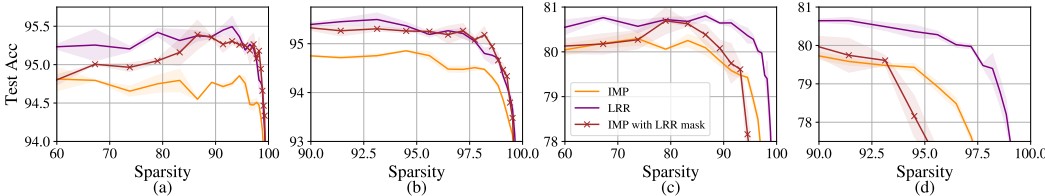

Figure 3: The sparse mask learnt by LRR is superior and the performance of IMP is improved in combination with the LRR mask on (a, b) CIFAR10 and (c, d) CIFAR100.

**LRR is more flexible in optimizing different masks.** According to Fig. 4(a, b), training LRR with the IMP mask (blue curve) is able to improve over IMP for CIFAR10. While the original LRR is less competitive for learning with the IMP mask on CIFAR100, LRR with BN parameter rewinding after each pruning iteration outperforms IMP both on CIFAR10 and CIFAR100 even at high sparsities. Similar results for Tiny ImageNet are presented in Fig. 2(d). Yet, are IMP and LRR masks sufficiently diverse? Since IMP and LRR masks are identified based on a similar magnitude based pruning criterion, the other mask and parameter initialization might still carry relevant information for the respective optimization task. In order to completely decouple the sparse mask from the parameter optimization and the initialization, we also study the LRR and IMP parameter optimization on a random mask.

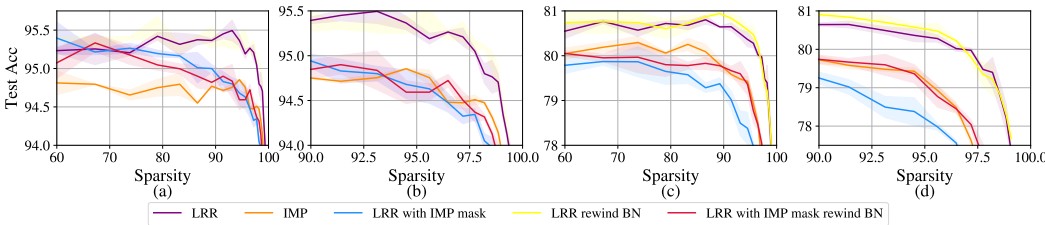

Figure 4: LRR improves parameter optimization within the mask learnt by IMP for (a, b) CIFAR10 and (c, d) CIFAR100.

**Random Masks.** For the same randomly pruned mask with balanced sparsity ratios (Gadhikar et al., 2023) and identical initialization, we compare training from initial values (IMP-rand) or training from the values obtained by the previous training–pruning iteration (LRR-rand) (see Fig. 2(b, c)). Rewinding the BN parameters assists gradual random pruning and improves optimization, thus, LRR-rand (rewind BN) outperforms IMP-rand. This confirms that LRR seems to employ a more flexible parameter optimization approach irrespective of task specific masks.

Our theoretical insights align with the observation that LRR learns network parameters more reliably than IMP. The main mechanism that strengthens LRR in our toy model is the fact that it inherits parameter signs that are identified by training an overparameterized model that is sufficiently flexible to correct initially problematic weight signs. To investigate whether a similar mechanism supports LRR also in a more complex setting, we study the sign flip dynamics.

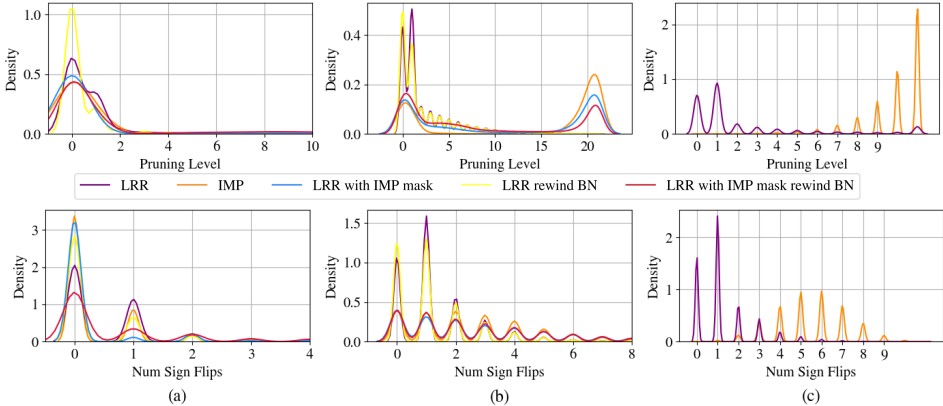

Figure 5: (top) The pruning iteration at which the parameter signs do not change anymore for LRR (purple) is much earlier than IMP (orange). (bottom) The number of times a parameter switches sign over pruning iterations (a) CIFAR10 (b) CIFAR100 and (c) Tiny ImageNet.

**LRR enables early and stable sign switches.** Fig. 4 confirms that LRR corrects initial signs primarily in earlier iterations when the mask is denser and the model more overparameterized. Moreover, the signs also stabilize early and remain largely constant for the subsequent pruning iterations (Fig. 5). Learnt parameters at consecutive sparsity levels in LRR tend to share the same sign in later iterations, but IMP must align initial signs in each pruning iteration, leading to unstable, back and forth flipping of learnt signs across sparsity levels. Overall, LRR changes more signs than IMP at lower sparsities on CIFAR10, yet, the effect is more pronounced in larger networks for CIFAR100 and Tiny ImageNet, where IMP fails to identify stable sign configurations even at high sparsities (see also Fig. 13 in appendix). These results apply to settings where the mask and parameter learning is coupled. Constraining both IMP and LRR to the same mask, LRR also appears to be more flexible and is able to improve performance by learning a larger fraction of parameter signs earlier than IMP (see Fig. 5(b)). For random masks, generally more unstable sign flips occur due to the fact that the mask and parameter values are not aligned well. Yet, LRR appears to be more stable and is able to flip more signs overall (Fig. 14 (a, b) in appendix). Even with the improved LRR mask, IMP seems unable to perform effective sign switches (Fig. 14 (c, d) in appendix).

Yet, maybe the LRR optimization can simply tolerate more sign switches? Furthermore, is LRR only able to switch signs in early training rounds due to the higher overparameterization of the networks? To answer these questions and learn more about the causal connection between sign switches and learning, next we study the effect of sign perturbations.

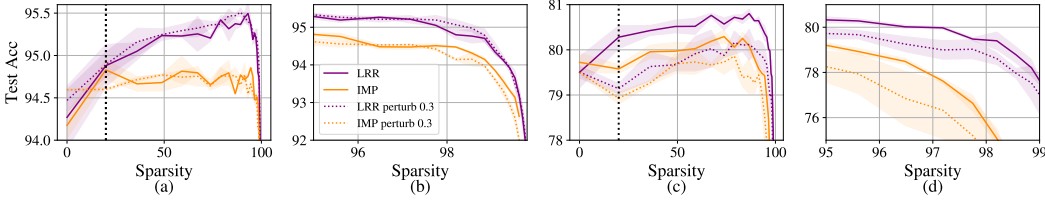

Figure 6: 30% signs in each layer are flipped randomly at 20% sparsity for LRR and IMP (dotted) on (a, b) CIFAR10 and (c, d) CIFAR100. Solid lines denote baselines.

**LRR recovers from random sign perturbations.** In order to characterize the effect of correct parameter signs on mask identification, we randomly perturb signs at different levels of sparsity

for both LRR and IMP. Sign perturbation at a low sparsity has little effect on CIFAR10 and both LRR and IMP are able to recover achieving baseline accuracy (Fig. 6(a, b)). For the more complex CIFAR100 dataset, signs have a stronger influence on masks and neither LRR nor IMP can fully recover to baseline performance. However, LRR is still able to achieve a higher performance than the IMP baseline, but IMP struggles after perturbing initial signs, as the mask does not fit to its initialization (Fig. 6(c, d)).

Fig. 7(a,b) shows results for perturbing a larger fraction of signs at much higher sparsity, i.e., 83%. LRR is able to recover over IMP at later sparsities on CIFAR10. Interestingly, on CIFAR100, LRR suffers more than IMP from the sign perturbation potentially due to a lack of overparameterization at high sparsity. LRR recovers slowly but still achieves baseline performance beyond 95% sparsity. The performance of subsequent masks obtained after perturbing signs reaffirms that parameter signs strongly influence the quality of the mask identification and LRR is capable of rearranging signs in order to find a better mask and optimize the corresponding parameters effectively. Yet, LRR requires training time and initial overparameterization to be effective.

**The interplay of magnitude and signs.** Recent analyses of IMP (Frankle et al., 2020b; Zhou et al., 2019) have found that signs that are learnt at later iterations are more informative and initializing with them improves IMP. In line with this insight, Fig. 7(c) highlights that rewinding only weight amplitude while maintaining the learnt signs improves over IMP. Yet, according to Frankle et al. (2020b) the combination with learned weight magnitudes can further strengthen the approach. Our next results imply that the magnitudes might be more relevant for the actual mask learning than the parameter optimization. We find that the learnt signs and the LRR mask contain most of the relevant information. Fig. 7(c) confirms that if we initialize IMP with the LRR signs and restrict it to the LRR mask, we can match the performance of LRR despite rewinding the weight magnitudes in every iteration. These results imply that a major drawback of IMP as a parameter optimization procedure could be that it forgets crucial sign information during weight rewinding.

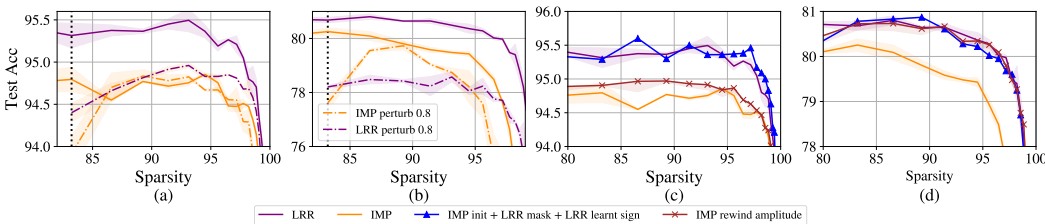

Figure 7: 80% signs in each layer are flipped randomly at 83% sparsity for LRR and IMP (dashed) on (a) CIFAR10 and (b) CIFAR100. Rewinding only magnitudes while using the initial weights of IMP with the learnt LRR masks and signs on (c) CIFAR10 and (d) CIFAR100.

## 4 CONCLUSIONS

Learning Rate Rewinding (LRR), Iterative Magnitude Pruning (IMP) and Weight Rewinding (WR) present cornerstones in our efforts to identify lottery tickets and sparsify neural networks, but the reasons for their successes and limitations are not well understood. To deepen our insights into their inner workings, we have highlighted a mechanism that gives LRR a competitive edge in structure learning and parameter optimization.

In a simplified single hidden neuron model, LRR provably recovers from initially problematic sign configurations by inheriting the signs from a trained overparameterized model, which is more robust to different initializations. This main theoretical insight also applies to more complex learning settings, as we show in experiments on standard benchmark data. Accordingly, LRR is more flexible in switching signs during early pruning–training iterations by utilizing the still available overparameterization. As a consequence, LRR identifies not only highly performant masks. More importantly, it can also optimize parameters effectively given diverse sets of masks. In future, we envision that insights into the underlying mechanisms like ours could inspire the development of more efficient sparse training algorithms that can optimize sparse networks from scratch.

ACKNOWLEDGEMENTS

We gratefully acknowledge funding from the European Research Council (ERC) under the Horizon Europe Framework Programme (HORIZON) for proposal number 101116395 SPARSE-ML.

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

## A  APPENDIX

### A.1  PROOFS: ONE DIMENSIONAL INPUT

**Theorem A.1.** *Let a target function $t(x) = \phi(x)$ and network $f(x) = a\phi(wx)$ be given such that $a$ and $w$ follow the gradient flow dynamics induced by Eq. (1) with a random balanced parameter initialization and sufficiently many samples. If $a(0) > 0$ and $w(0) > 0$, $f(x)$ can learn the correct target. In all other cases $(a(0) > 0, w(0) < 0)$, $(a(0) < 0, w(0) > 0)$ and $(a(0) < 0, w(0) < 0)$ learning fails.*

*Proof:* The derivatives of the parameters $(a, w)$ with respect to the loss are given by

$$\frac{\partial \mathcal{L}}{\partial a} = -\frac{1}{n} \sum_{i=1}^{n} (y_i - a\phi(wx_i)) \phi(wx_i); \quad \frac{\partial \mathcal{L}}{\partial w} = -\frac{1}{n} \sum_{i=1}^{n} (y_i - a\phi(wx_i)) ax_i \mathbb{1}_{(wx_i > 0)}, \quad (2)$$

which induce the following ODEs under gradient flow:

$$\dot{a} = -aw^2 C_1 + wC_2, \quad \dot{w} = -wa^2 C_1 + aC_2, \quad (3)$$

where $C_1 = \frac{1}{n} \sum_{i \in \mathcal{I}} x_i^2$ and $C_2 = \frac{1}{n} \sum_{i \in \mathcal{I}} (x_i \phi(x_i) + \zeta_i x_i)$. Note that $C_1$ and $C_2$ can change over time, as they depend on the potentially changing set $\mathcal{I}(t)$ that comprises all samples on which the neuron is switched on: $\mathcal{I}(t) := \{i | w_i(t)x_i > 0\}$.

To solve this set of ODEs, we have to dinstinguish multiple cases. For all of them, we have $a^2(t) = w^2(t)$. Thus, both parameters can pass through zero only together and potentially switch their sign only in this case. If $w(t)$ does not switch its sign during training, $C_1$ and $C_2$ remain constant, we know that $a(t) = \text{sign}(a(0))|w(t)|$ and we can focus on solving the dynamic equation for $w$ at least until a potential sign switch. Replacing $a(t) = \text{sign}(a(0))|w(t)|$ in the ODE for $w$ leads to $\dot{w} = -w^3 C_1 + w\tilde{C}_2$ with $\tilde{C}_2 = C_2 \text{sign}(a(0)) \text{sign}(w(0))$. As this ODE is of Bernoulli form, it has a closed form solution. Note that generally $C_1 \geq 0$. If $C_1 = 0$, nothing happens and our function remains a constant 0. Otherwise, we have

$$w(t) = \frac{\sqrt{\tilde{C}_2} w(0) \exp(2\tilde{C}_2 t)}{\sqrt{\tilde{C}_2 - C_1 w(0)^2} \sqrt{\frac{C_1 w(0)^2 \exp(4\tilde{C}_2 t)}{\tilde{C}_2 - C_1 w(0)^2} + 1}}, \text{ if } \tilde{C}_2 > 0, \quad (4)$$

$$w(t) = \frac{\text{sign}(w(0))\sqrt{-\tilde{C}_2}}{\exp\left(-2\tilde{C}_2 t\right)\left(C_1 - \tilde{C}_2/w^2(0)\right) - C_1}, \text{ if } \tilde{C}_2 < 0, \text{ and} \quad (5)$$

$$w(t) = \frac{w(0)}{\sqrt{2C_1 w^2(0)t + 1}} \text{ if } \tilde{C}_2 = 0. \quad (6)$$

which can be easily verified by differentiation. Note that these equations only hold until $w(t)$ passes through 0. If that happens at $t_0$, $C_1$ and $C_2$ actually change and we have to modify the above equations. Technically, if $w(t_0) = 0$, the gradient flow dynamics halt because $C_1(t_0) = 0$ and $C_2(t_0) = 0$. Yet, in a noisy learning setting with discrete step sizes, it is not impossible that parameters switch their sign. In this case, a new dynamics start from the switching point $t_0$ on a time scale $\tilde{t} = t - t_0 - \epsilon$ that continues from the previously found parameters $w(\tilde{t} = 0) = w(t = t_0 + \epsilon)$. We now develop an intuition of what this means for our learning problem, by differentiating the problem into different cases based on initial parameter signs.

**Correct initial sign.** If we start with the correct signs $w(0) > 0$ and $a(0) > 0$ as the target, then our neuron has no problem to learn the right parameters given enough samples. $w(0) > 0$ implies that $\mathcal{I} = \{i | x_i > 0\}$ so that $C_2 = 1/n \sum_{i \in \mathcal{I}} (x_i \phi(x_i) + \zeta_i x_i) = 1/n \sum_{i \in \mathcal{I}} x_i^2 + 1/n \sum_{i \in \mathcal{I}} \zeta_i x_i$. According to the law of large numbers, $\lim_{n \to \infty} C_2 = \mathbb{E}(\phi(X)_1^2) + \mathbb{E}(\phi(X)_1 \zeta) = 1/(2d) > 0$ and $\lim_{n \to \infty} C_1 = \mathbb{E}(\phi(X)_1^2) = 1/(2d)$ almost surely. Also for finite samples, we likely have $C_2 > 0$, as we will make more precise later. Thus, $\tilde{C}_2 = \text{sign}(w(0)a(0))C_2 = C_2 > 0$ and $w$ converges to $w(\infty) = \sqrt{\tilde{C}_2/C_1} = 1$ without passing through 0. Also $a(\infty) = \text{sign}(a(0))|w(\infty)| = 1$ corresponds to the correct target value.

**Case** $w(0) > 0$ **but** $a(0) < 0$**.** Since $\mathcal{I}$ is defined as above, our initial constants $C_1$ and $C_2$ are identical. Yet, $\tilde{C}_2 = -C_2 < 0$ changes its sign, which has a considerable impact on the learning dynamics. As $a(0)$ has started with the wrong sign, the gradient dynamics try to rectify it and send $w(t)$ to $0$ in the process, as $a(t)$ would need to pass through $0$ to switch its sign. Thus, learning fails. In case of finite samples, $\tilde{C}_2$ is subject to noise. If $\tilde{C}_2 < 0$, $a$ and $w$ still converge to $0$. However, if $\tilde{C}_2 > 0$ because of substantial noise, $w$ would converge to a positive value $w(\infty) = \sqrt{\tilde{C}_2/C_1}$, while $a(\infty) = -\sqrt{\tilde{C}_2/C_1} < 0$, which would not align at all with the target $\phi(x)$.

**Case** $w(0) < 0$**.** This case is bound to fail regardless of the sign of $a(0)$, if the noise is not helping with a sign switch. The reason is that the neuron is initially switched on only on the negative samples $\mathcal{I} = \{i | x_i < 0\}$, for which the true labels are zero. In consequence, $C_2 = 1/n \sum_{i \in \mathcal{I}} (x_i \phi(x_i) + \zeta_i x_i) = 1/n \sum_{i \in \mathcal{I}} \zeta_i x_i$ and $\lim_{n \to \infty} C_2 = \mathbb{E}(-\phi(-X)_1 \zeta) = 0$ almost surely. Thus, the training data provides an incentive for $a$ and $w$ to converge to $0$ without passing through $0$ in between. Also for finite samples, we have $\tilde{C}_2 = -a(0)/n \sum_{i \in \mathcal{I}} \zeta_i x_i > 0$ with probability $1/2$. In this case, $w$ will converge to a negative value $\text{sign}(w_0)\sqrt{\tilde{C}_2/C_1}$. If $\tilde{C}_2 < 0$, then both $w$ and $a$ converge to $0$ without the opportunity to change the sign with a too large discrete gradient step.

**Finite sample considerations.** Interestingly, the number of samples and the noise level do not really influence the (failed) learning outcome in case of a negative initial weight $w(0) < 0$. The case $w(0) > 0$, $a(0) < 0$ is not able to learn a model that can come close to the ground truth target. Thus, only the potential success case $w(0) > 0$ and $a(0) > 0$ depends meaningfully on the data and its signal to noise ratio, as our set-up reduces to an overparameterized linear regression problem with outcome $a(\infty) = w(\infty) = \sqrt{C_2/C_1}$ if $C_2 > 0$. (Note that $C_2 < 0$ would imply such high noise that the learning could also not be regarded successful, as $a(\infty) = w(\infty) = 0$.) The sample complexity of learning the parameters is the only part that depends on distribution assumptions regarding the input and noise. The effective regression parameter $a(\infty)w(\infty) = C_2/C_1 = 1 + (\sum_{i \in \mathcal{I}} \zeta_i x_i)/(\sum_{i \in \mathcal{I}} x_i^2)$ depends in the usual way on the noise, but requires double the number of samples as a normal regression problem, as in approximately half of the cases, the neuron is switched off.

## A.2 Proofs: Overparameterized Input ($d > 1$)

In the following section, we prove our main theorem that allows us to conclude that LRR has a higher chance to succeed in learning a single univariate neuron than IMP.

Learning an overparameterized multivariate neuron $f(\boldsymbol{x}) = a\phi(\boldsymbol{w}^T \boldsymbol{x})$ for $\boldsymbol{x} \in \mathbb{R}^d$ corresponds to a more complex set of coupled gradient flow ODEs, if $d > 1$.

$$\dot{\boldsymbol{w}} = -a^2 \mathbf{C_1} \boldsymbol{w} + a \mathbf{C_2}, \quad \dot{a} = -a \boldsymbol{w}^T \mathbf{C_1} \boldsymbol{w} + \mathbf{C_2}^T \boldsymbol{w},$$

$$\text{with } \mathbf{C_1} = \frac{1}{n} \sum_{i \in \mathcal{I}} \mathbf{x_i} \mathbf{x_i}^T, \quad \mathbf{C_2} = \frac{1}{n} \sum_{i \in \mathcal{I}} y_i \mathbf{x_i}, \tag{7}$$

where the dynamic set $\mathcal{I}$ is again defined as the set of samples on which the neuron is activated so that $\mathcal{I} = \{i \mid \boldsymbol{w}_i^T \boldsymbol{x}_i > 0\}$. The main difference to the previous one-dimensional case is that this set is initially not determined by $w_1(0)$. Even in case of a problematic initialization $w_1(0) < 0$, the neuron can learn a better model because of $c_{2,1} > 0$.

We cannot expect to derive the gradient flow dynamics for this problem in closed form, as $\mathbf{C_1}$ and $\mathbf{C_2}$ depend on $\boldsymbol{w}$ in complicated nonlinear ways. However, the structure of a solution is apparent, as the problem corresponds to an overparameterized linear regression problem given $\mathcal{I}$. Lee et al. (2022a) have discussed the solutions to this general problem in case of positive input and fixed, non-trainable $a$. Assuming balancedness $a^2 = \|\boldsymbol{w}\|^2$, our solution must also be of the form $\boldsymbol{w} = \beta/\sqrt{\|\beta\|}$ and $a = \text{sign}(a)\sqrt{\|\beta\|}$, where $\beta = \mathbf{X}^+ \boldsymbol{y}$ is the mean squared error minimizer and $\mathbf{X} = (x_{i,k})_{ik} \in \mathbb{R}^{|I| \times d}$ corresponds to the data matrix on the active samples. Under conditions that enable successful optimization, we obtain $\beta \approx \boldsymbol{v} = (1, 0, \ldots)^T$.

Yet, there are still several issues that can arise during training as the set $I$ changes with $\boldsymbol{w}$. Solving the set of ODEs is generally a hard problem, even though several variants have been well studied

Yehudai & Ohad (2020); Lee et al. (2022a); Vardi et al. (2021;?); Oymak & Soltanolkotabi (2019); Soltanolkotabi (2017); Kalan et al. (2019); Frei et al. (2020); Diakonikolas et al. (2020); Tan & Vershynin (2019); Du et al.. Most works assume that the outer weight is fixed $a(0) = a(t) = 1$ and only the inner weight vector $\boldsymbol{w}$ is learned Yehudai & Ohad (2020); Lee et al. (2022a); Vardi et al. (2021). Only (Boursier et al., 2022) considers trainable $a$ but excludes the single hidden neuron case and is restricted to orthogonal inputs. Furthermore, the noise is usually considered to be $0$. In the large sample setting, this assumption would be well justified, as the noise contribution to $\mathbf{C_2}$ approaches $0$. However, to guarantee convergence, additional assumptions on the data generating process are still required, as Yehudai & Ohad (2020) have pointed out with a counter example that for a given parameter initialization method (in form of a product distribution) there exist a data distribution for which gradient flow (or SGD or GD) does not converge with probability at least $1 - \exp(-d/4)$. Vardi et al. (2021) have furthermore shown that if we also want to learn biases (which would be the case if one of our data input components is constant $1$), a uniform initial weight distribution could lead to failed learning results with probability close to $1/2$.

Recall that three scenarios prevent IMP from succeeding in learning our target $\phi(x_1)$: a) $a(0) < 0$, $w_1(0) > 0$, b) $a(0) < 0$, $w_1(0) < 0$, and (c) $a(0) > 0$ and $w_1(0) < 0$. LRR cannot succeed in case of (a) and (b) as well, because $a$ cannot switch its sign to $a(\infty) > 0$ during training, as the following lemma states.

**Statement** (Restated Lemma 2.2). *Assume that $a$ and $\boldsymbol{w}$ follow the gradient flow dynamics induced by Eq. (7) with Gaussian iid input data, zero noise, and that initially $0 < |a|\|\boldsymbol{w}(0)\| \leq 2$ and $a(0)^2 = \|\boldsymbol{w}(0)\|^2$. Then $a$ cannot switch its sign during gradient flow.*

*Proof.* This statement follows immediately from the balancedness property. To switch its sign, $a(t)$ would need to pass through zero. Thus, let us assume there exists a time point $t_0 > 0$ so that $a(t_0) = 0$. Since $a(t)^2 = \|\boldsymbol{w}(t)\|^2$ for the complete dynamics, this implies that $\|\boldsymbol{w}(t_0)\|^2 = 0$. As this switches of the neuron, $\mathbf{C_1(t_0)} = \mathbf{C_2(t_0)} = \mathbf{0}$ so that $\dot{a}(t_0) = 0$ and $\dot{w}_k(t_0) = 0$. It follows that $a(t) = 0$ and $\|\boldsymbol{w}(t)\|^2 = 0$ for all $t \geq t_0$ so that no sign switch occurs. $\square$

Note that for finite, relatively high learning rates, it could be possible that a neuron switches its sign because it never switches off completely and instead overshoots $0$ with a large enough gradient step. In most cases, this would provide LRR with an advantage. Because if a problematic initialization with $a(0) < 0$ could be mitigated by training so that $a(\infty) > 0$, LRR would benefit but not IMP. The only scenario in favor for IMP would be the case that the initialization is advantageous so that $a(0) > 0$ and $w_1(0) > 0$ but becomes problematic during training so that $a(\infty) < 0$. This, however, would require such high noise that also training an univariate neuron from scratch could not result in a good model. It is therefore an irrelevant (and unlikely) case and does not impact our main conclusions, which are restated below for convenience.

**Statement** (Restated Theorem 2.3). *Assume that $a$ and $\boldsymbol{w}$ follow the gradient flow dynamics induced by Eq. (7) with Gaussian iid input data, zero noise, and that initially $0 < |a|\|\boldsymbol{w}(0)\| \leq 2$ and $a(0)^2 = \|\boldsymbol{w}(0)\|^2$. If $w_1(0) < 0$ and $a(0) > 0$, LRR attains a lower objective (1) than IMP. In all other cases, LRR performs at least as well as IMP.*

*Proof.* According to Lemma 2.2, $a(0) \leq 0$ implies $a(\infty) \leq 0$ so that neither IMP nor LRR can succeed to learn the correct univariate target neuron after pruning. We can therefore focus our analysis on the case $a(\infty) > 0$. Note that this implies that $a(t) = \|\boldsymbol{w}(t)\|$ because $a$ does not switch its sign.

Both IMP and LRR rely on the first overparameterized training cycle to result in a successful mask identification, which requires $|w_1| >> |w_i|$ for any $i \neq 1$. Otherwise, both approaches (IMP and LRR) would fail. Hypothetically, it could be possible that $|w_1(\infty)| >> |w_i(\infty)|$ while $w_1(\infty) < 0$, which would not correspond to a successful training round, since $\boldsymbol{w}(\infty) \neq (1, 0, 0, ...)$ but would result in a correct mask identification. This case would be interesting, as it would allow IMP to succeed if $w_1(0) > 0$ while LRR could not, as it would start training an univariate neuron from $w_1(\infty) < 0$. However, note that the derivative of $\boldsymbol{w}$ would be nonzero in this case. Thus, there exists no stationary point with the property $w_1(\infty) < 0$.

In consequence, only cases of successful training offer interesting instances to compare IMP and LRR. For our argument, it would be sufficient to show that learning a multivariate neuron is suc-

cessful with nonzero probability and argue that LRR succeeds while IMP fails in some of these cases. Yet, we can derive a much stronger statement by using and adapting Theorem 6.4 by Yehudai and Shamir (Yehudai & Ohad, 2020) and prove that learning is generally successful for reasonable initializations.

**Learning a single neuron.** We define $z(t) = a(t)w(t)$. Note that $z$ has therefore norm $\|z\| = \|w\|^2$ and its direction coincides with the one of $w$, as $z/\|z\| = w/\|w\|$. In case of successful training, we expect $z(t) \to v$ for $t \to \infty$, where $v$ is a general target vector. In our case, we assume $v = (1, 0, 0, ...)$. Our main goal is to bound the time derivative $\delta/\delta t \|z(t) - v\|^2 = 2 \langle \dot{z}(t), z(t) - v \rangle \leq -\lambda \|z(t) - v\|^2$ for a $\lambda > 0$. Grönwall's inequality would then imply that $\|z(t) - v\|^2 \leq \|z(0) - v(0)\|^2 \exp(-\lambda t)$ and hence $z(t) \to v$ exponentially fast.

In contrast to (Yehudai & Ohad, 2020), the derivative $\dot{z}(t) = a\dot{w} + \dot{a}w$ consists of two parts that we have to control separately.

It is easy to see based on Eq. (7) that the time derivative of $w(t)$ fulfills:

$$\langle a\dot{w}, z - v \rangle = -a^2 \frac{1}{n} \sum_{i \in I, v^T x_i > 0} \langle x_i, z - v \rangle^2 \tag{8}$$

$$-a^2 \frac{1}{n} \sum_{i \in I, v^T x_i \leq 0} \langle x_i, z - v \rangle \langle x_i, z \rangle \tag{9}$$

$$= -a^2 \|z - v\|^2 \frac{1}{n} \sum_{i \in I, v^T x_i > 0} \|x_i\|^2 \cos(\varphi(x_i, z - v))^2 \tag{10}$$

$$-a^2 \frac{1}{n} \sum_{i \in I, v^T x_i \leq 0} \langle x_i, z \rangle^2 - a^2 \frac{1}{n} \sum_{i \in I, v^T x_i \leq 0} \langle x_i, -v \rangle \tag{11}$$

$$\leq -a^2 \|z - v\|^2 \lambda_1 = -\lambda_1 \|z\| \|z - v\|^2 \geq -\lambda_0 \|z - v\|^2, \tag{12}$$

where we dropped the term (11) because it is negative. (Note that all involved factors are positive because $-v^T x_i \geq 0$.) Furthermore, we have $\lambda_1 = 1/n \sum_{i \in I, v^T x_i > 0} \|x_i\|^2 \cos(\varphi(x_i, z - v))^2 > 0$ with high probability with respect to the data distribution according to Lemma B1 by Yehudai and Shamir (Yehudai & Ohad, 2020). Similarly, the proof of Thm. 5.3 by (Yehudai & Ohad, 2020) argues why $a^2 = \|z\| > 0$ is bounded from below, which allows us to integrate its lower bound into the constant $\lambda_0$.

The second term of $\dot{z}$ is not considered by (Yehudai & Ohad, 2020), as they assume that $a$ is not trainable. We get:

$$\langle \dot{a}w, z - v \rangle = \dot{a} \langle w, z - v \rangle \leq 0. \tag{13}$$

The last inequality can be deduced by distinguishing two cases. If $\langle w, z - v \rangle > 0$, then $\dot{a} < 0$. If $\langle w, z - v \rangle < 0$, then $\dot{a} > 0$. This follows from the fact that

$$\dot{a} = -\|w\|^3 \frac{1}{n} \sum_{i \in I} \left\langle \frac{w}{\|w\|}, x_i \right\rangle + \frac{1}{n} \sum_{i \in I, v^T x_i > 0} \langle v, x_i \rangle \langle w, x_i \rangle \tag{14}$$

and that $\langle w, z - v \rangle = \|w\|^3 - \langle w, v \rangle = \|w\|^3 - w_1$. On average with respect to the data, we thus receive Eq. (13). Note that the normal case is that $\langle w, z - v \rangle > 0$, as this holds initially with high probability and it remains intact during training.

Combining Eq. (12) and Eq. (13) completes our argument, since

$$\frac{\delta \|z(t) - v\|^2}{\delta t} = 2 \langle \dot{z}(t), z(t) - v \rangle \leq -\lambda \|z(t) - v\|^2 \tag{15}$$

for a $\lambda > 0$. Grönwall's inequality leads to $\|z(t) - v\|^2 \leq \|z(0) - v(0)\|^2 \exp(-\lambda t)$ and hence $z(t) \to v$ exponentially fast.

**LRR outperforms IMP.** If training the overparameterized neuron is successful with $a(0) > 0$ and $a(\infty) > 0$ as discussed previously, then $w_1^{(1)}(\infty) = 1 > 0$. After pruning, LRR has to train an

univariate neuron with the initial condition $w_1^{(2)}(0) = w_1^{(1)}(\infty) > 0$, which converges to the correct ground truth model. However, if $w_1^{(1)}(0) < 0$, IMP has to start training from $w_1^{(2)}(0) = w_1^{(1)}(0) < 0$, which leads to a wrong model estimate, as discussed in Section 2.1.

$\square$

## A.3 EXPERIMENTAL DETAILS

| Dataset | CIFAR10 | CIFAR100 | Tiny ImageNet | ImageNet |
|---|---|---|---|---|
| Model | ResNet18 | ResNet50 | ResNet50 | ResNet50 |
| Epochs | 150 | 150 | 150 | 90 |
| LR | 0.1 | 0.1 | 0.1 | 0.1 |
| Scheduler | cosine-warmup | step-warmup | step-warmup | step-warmup |
| Batch Size | 256 | 256 | 256 | 256 |
| Warmup Epochs | 50 | 10 | 50 | 10 |
| Optimizer | SGD | SGD | SGD | SGD |
| Weight Decay | 1e-4 | 1e-3 | 1e-3 | 1e-4 |
| Momentum | 0.9 | 0.9 | 0.9 | 0.9 |
| Init | Kaiming Normal | Kaiming Normal | Kaiming Normal | Kaiming Normal |

Table 1: Experimental Setup

**Image Classification Tasks.** Table 1 details our experimental setup. In each pruning iteration, we keep $80\%$ of the currently remaining parameters of highest magnitude (Frankle & Carbin, 2019).

**Warmup Epochs.** Each training run of a dense network starts with warmup epochs with a linearly increasing learning rate from upto $0.1$. Weights are rewound to their values after warmup in case of IMP (i.e. similar to WR). We ensure that each run of IMP and LRR has an identical rewind point (after warmup epochs).

The learning rate schedules we found to achieve the best performance in our experiments as confirmed in Figure 8 and 9:

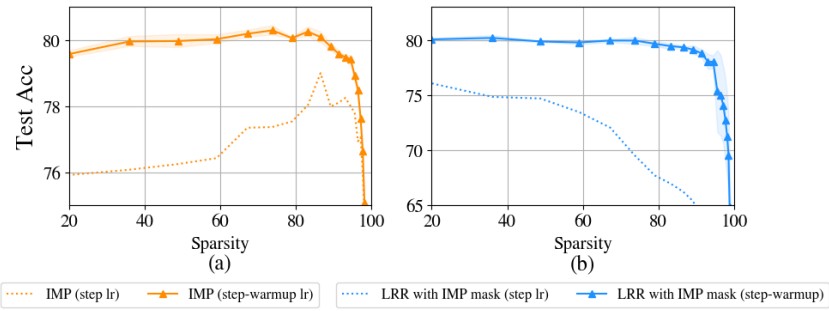

Figure 8: Comparing LR schedules on CIFAR100.

**(i) cosine-warmup**: Learning rate is increased linearly to $0.1$ for the first $10$ epochs, followed by a cosine lr schedule for the remaining $140$ epochs.

**(ii) step-warmup**: Learning rate is increased linearly to $0.1$ for the first $10$ epochs, followed by a step lr schedule for the remaining $140$ epochs with learning multiplied by a factor of $0.1$ at epochs $60$ and $120$.

**(iii) ImageNet:** For ImageNet, we use a constant learning rate of $0.01$ during the warmup phase. The learning rate is reduced by a factor of $10$ every $30$ epochs after the warmup phase, starting from $0.1$.

In case of random pruning we randomly remove parameters in each layer in order to maintain a balanced layerwise sparsity ratio (Gadhikar et al., 2023) i.e. every layer has an equal number of

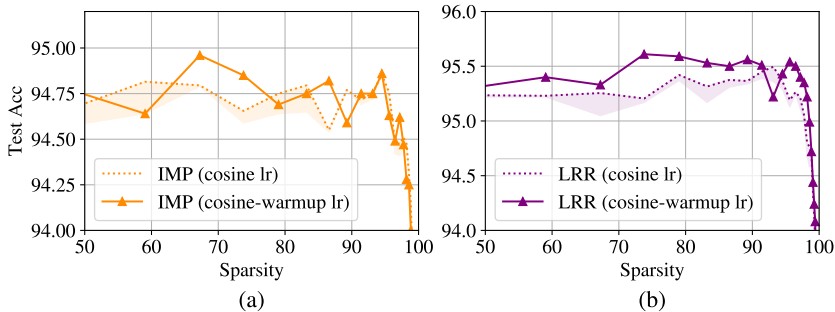

Figure 9: Comparing LR schedules on CIFAR10.

nonzero parameters. Each run is repeated thrice and we report the mean and 95% confidence interval of these runs. All experiments are performed on a Nvidia A100 GPU. Our code is built on the work of (Kusupati et al., 2020).

## A.4 ADDITIONAL RESULTS

**BN parameter distributions**: We plot the layer wise distributions of the learnable scaling parameter $\gamma$ for our experiments. The distributions are similar on CIFAR10 (Figure 10) with most values being positive in every layer for IMP, LRR and LRR with IMP mask. Hence, rewinding BN parameters also finds similar distributions.

However, the rewinding BN improves performance of LRR with IMP mask on CIFAR100 (See Fig. 4). This can be attributed to the reducing the number of neurons (channels) where $\gamma = 0$ (eg: Layer 22, 23 in CIFAR100 Fig. 11). We find that this is is necessary in deeper layers to aid signal propagation and hence we propose rewinding BN parameters to aid LRR when the mask is decoupled from the optimization.

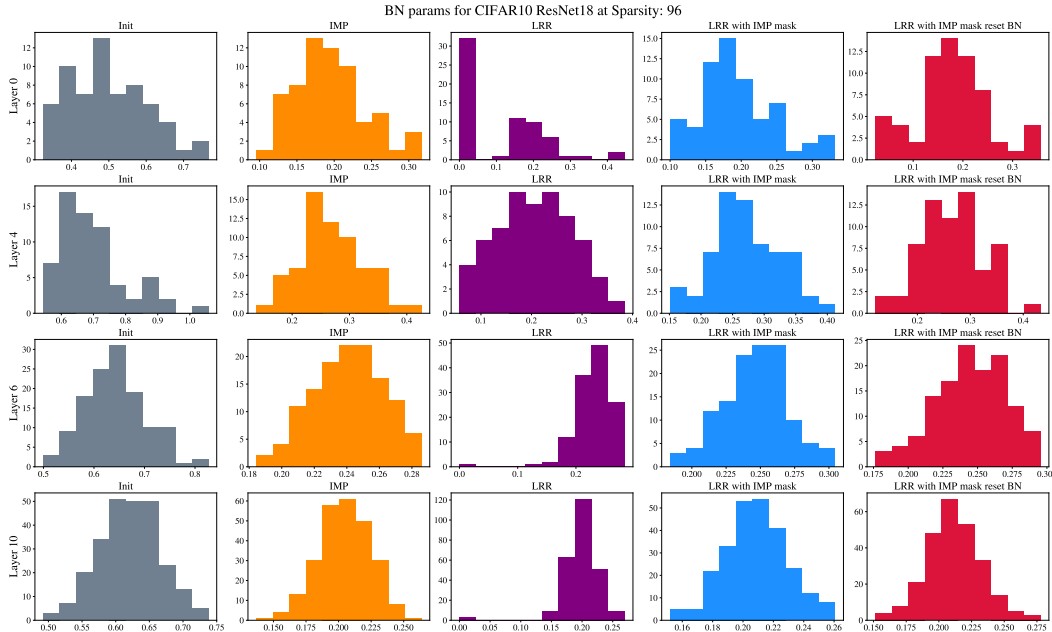

Figure 10: Layer wise distributions of BN parameter ($\gamma$) on CIFAR10 ResNet18 at Sparsity 96%.

**Interplay of magnitude and signs on CIFAR100.** On CIFAR100 too, the signs and mask learnt by LRR contain majority of the information required while training. When using the mask and signs

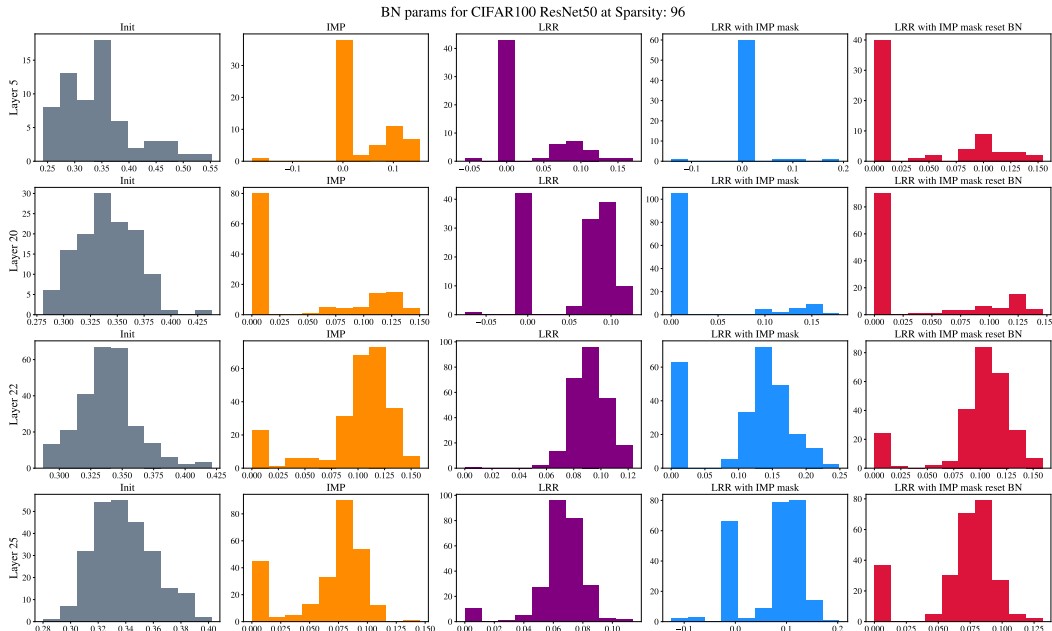

Figure 11: Layer wise distributions of BN parameter ($\gamma$) on CIFAR100 ResNet50 at Sparsity 96%.

learnt by LRR and only rewinding weight magnitudes, we match the performance of LRR (see Fig. 12).

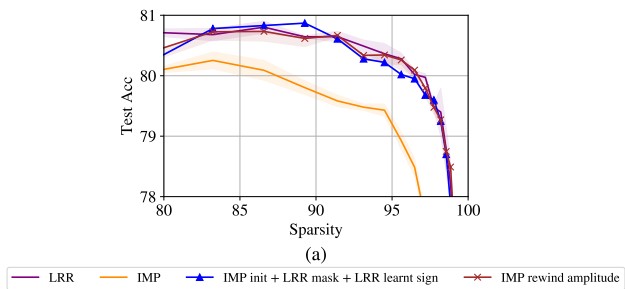

Figure 12: (a) For CIFAR100 we compare rewinding only magnitudes to using the initial weights of IMP with the learnt LRR masks and signs.

**Overall sign flips for LRR vs IMP.** As verified experimentally, LRR enables more sign flips early in training. This can also be confirmed by measuring the total number of sign flips from initial signs to learnt signs at each sparsity level for both LRR and IMP. We plot the difference between the number of sign flips enabled by LRR and IMP at each pruning iteration in Figure 13. Early sparsities show a large positive difference between the number of signs flipped by LRR and IMP, showing the ability of LRR to enable more sign flips.

**LRR enables early sign switches for parameter optimization.** Figure 14(a, b) shows for a randomized mask, LRR-rand enables a larger fraction of signs to switch earlier in training than IMP-rand in spite of unstable sign flips due to a randomized mask which does not align with parameter values. On the other hand, the ability to switch signs still lacks in IMP in spite of training with an improved LRR mask as shown in Figure 14(c, d) highlighting that the weight rewinding step leads to loss of sign information.

**LRR enables early sign switches on ImageNet.** We report results on ImageNet in Figure 15 (a) for a ResNet50. We find that our insights translate to the large scale setting as well. To support our hypothesis of the importance of learnt signs, we show that using the initial weight magnitudes of

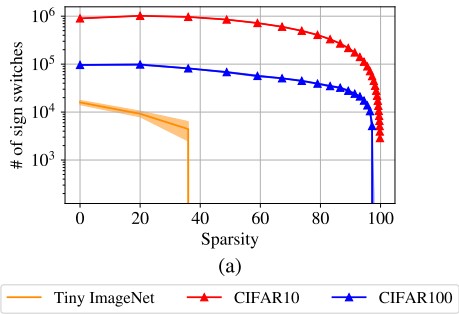

Figure 13: (a) Difference between number of sign slips between initial weights and weights at each sparsity level enabled by LRR and IMP. Positive differences indicate that LRR enables more sign flips than IMP.

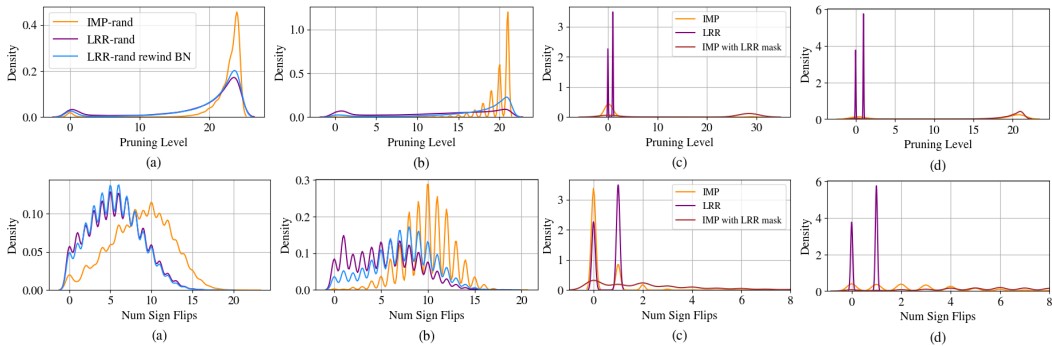

Figure 14: (top) The pruning iteration at which the parameter signs do not change anymore and (bottom) the number of times a parameter switches sign over pruning iterations, with a random mask for (a) CIFAR10 and (b) CIFAR100 and with an LRR mask for (c) CIFAR10 and (d) CIFAR100.

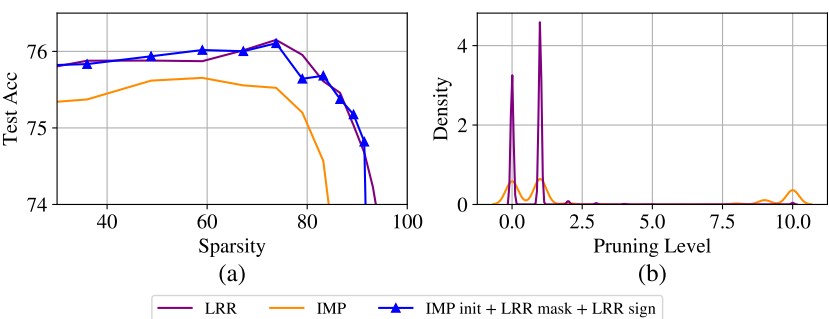

Figure 15: (a) Comparison of LRR vs IMP for a ResNet50 on ImageNet . (b) Histograms of the pruning iteration at which the parameter signs do not change anymore for ImageNet.

IMP with the mask learnt by LRR and the weight signs learnt by LRR, we are still able to match the performance of LRR (blue curve). Figure 15 (b) confirms that LRR enables early sign switches by the third prune-train iteration while IMP struggles to perform sign switches.

**Impact of pruning criteria.** In order to highlight that different pruning criteria can benefit from initial overparameterization with continued training in comparison with weight rewinding, we report results for pruning iteratively with Synflow (Tanaka et al., 2020) and SNIP (Lee et al., 2019). Although, these criteria have been proposed for pruning at initialization, we use them iteratively following the same prune-train procedure as LRR and IMP. Although LRR and IMP are used in the context of magnitude pruning, we use the same terms followed by the pruning criterion to differen-

tiate between training with learning rate rewinding (LRR (synflow/snip)) and training with weight rewinding (IMP (synflow/snip)). For eg: IMP (synflow) indicates that each prune - train iteration prunes weights based on the Synflow criterion and rewinds the nonzero weights back to their initial values.

Figure 16 supports our hypothesis that for different pruning criteria, like Synflow and SNIP, continued training benefits from initial overparameterization by enabling better sign switchyoes. The blue curve confirms that as long as the mask and sign learnt by LRR (synflow/snip) is used, we can match the performance of LRR (synflow/snip) denoted by the purple curve for any weight magnitude ((IMP init + LRR mask + LRR sign (snip/synflow)).

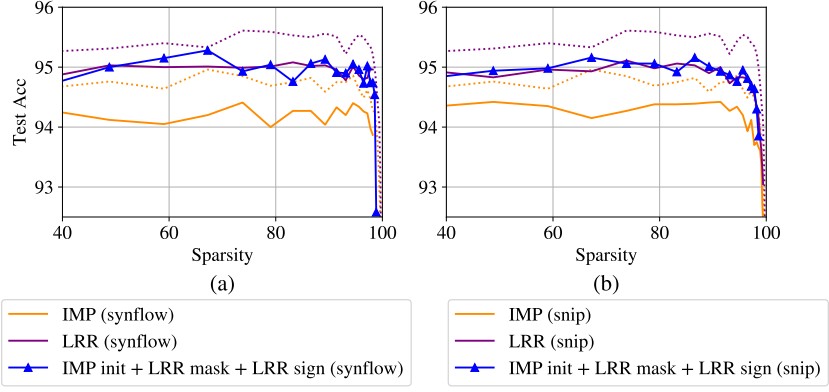

Figure 16: Effect of pruning criteria (a) Synflow and (b) SNIP with iterative pruning. LRR (synflow/snip) denotes iterative pruning with learning rate rewinding after every pruning step and IMP (synflow/snip) denotes iterative pruning with weight rewinding after every pruning step with the respective pruning criterion for a ResNet18 on CIFAR10. Dotted lines indicate baseline results of IMP and LRR using magnitude as the pruning criteria.

**CIFAR10 on VGG16.** We also report results for a VGG16 model with Batch Normalization (Simonyan & Zisserman, 2015) on CIFAR10. Figure 17(a) shows that LRR improves over IMP and that its signs and mask contain sufficient information to match the performance of LRR (see blue curve). Figure 17(b) further confirms that LRR enables early sign switches compared to IMP, supporting our hypothesis that LRR gains performance due to its ability to effectively switch signs in early iterations.

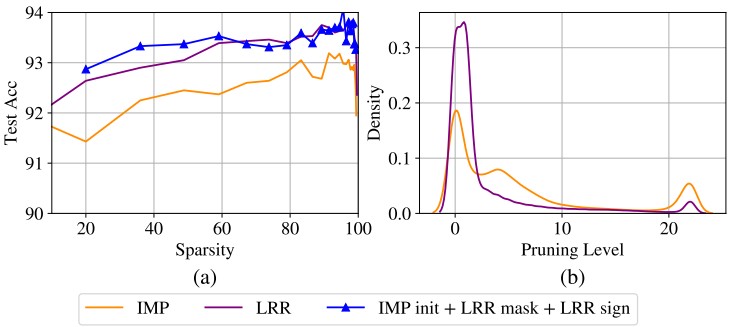

Figure 17: (a) Comparison of LRR versus IMP for a VGG16 model on CIFAR10. (b) Histogram of the pruning iteration at which the parameter signs do not change anymore.

**Impact of overparameterization on single hidden neuron network.** We show that increasing overparameterization in the input dimension $d$ for the single hidden neuron network defined in Section 2 aids LRR. We follow the same experimental setup as Section 2.3. In the case when $d = 1$, LRR and IMP are equally likely to succeed for the case where $a(0) > 0, w_1(0) > 0$ (see Figure 18(a)). If we increase $d > 1$, we find that LRR is now able to leverage the overparameterized model

in the early pruning iterations to flip initial bad signs and succeed more often than IMP (see Figure 18(b), (c), (d)) as long as $a(0) > 0$.

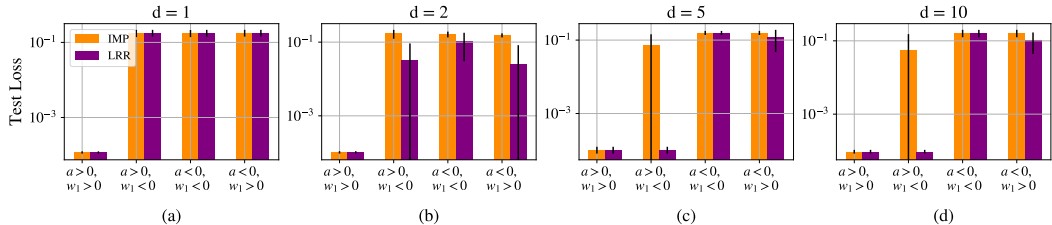

(a)          (b)          (c)          (d)

Figure 18: Effect of increasing input overparameterization (measured by the input dimension ($d$) for the single hidden neuron network. Without overparameterization (a) $d = 1$ and with overparameterization (b) $d = 2$ (c) $d = 5$ (d) $d = 10$.

