# OpenReview forum: "Masks, Signs, And Learning Rate Rewinding"
_ICLR.cc/2024/Conference — ICLR 2024 spotlight_

### Official Review · Reviewer_yLRG · 2023-10-30

**Soundness:** 3 good
**Presentation:** 3 good
**Contribution:** 3 good
**Rating:** 6
**Confidence:** 3

**Summary:**

The paper explores LRR and IMP, the key methods for identifying lottery tickets in large neural networks, with the goal to understand the differences between mask learning and parameter optimization. The paper provides valuable theoretical results for one hidden neuron networks predicting the LRR's superior performance due to its ability to overcome initial parameter sign challenges. Experiments with ResNets on CIFAR10, CIFAR100 and Tiny ImageNet demonstrate LRR's superior performance due to its ability to adjust the parameter signs early in training.

**Strengths:**

* The paper addresses and important, timely and high-impact problem, which could help to drastically improve the efficiency and decrease the cost of sparse training.
* Decoupling structure learning and parameter learning is interesting and the steps taken in the paper are meaningful.
* The theoretical results on one hidden neuron networks are interesting, strong, and well described.
* The experimental results support the claims of the paper, although the performance differences between LRR and IMP are small.

**Weaknesses:**

* Although the main ideas explored in the paper and the theoretical insights are strong, the empirical evaluation is limited to ResNet architectures. The empirical results are mainly reported for CIFAR10 and CIFAR100. Extending the findings to more architectures and other domains would be helpful to understand the significance of the findings. Additionally, to reviewer's understanding, several figures do not fully support the claims made in the text (see questions below).
* Magnitude pruning is explored as the only strategy to train sparse networks. Can the findings regarding the superiority of LRR be generalized and extended to other pruning strategies?
* An exploration of the impact of the learning rate would also help to better understand the practical value of the proposed analysis.

Minor: A missing reference [?] on p.3. "LRR is improves parameter optimization" on p.7.

**Questions:**

* The conclusions for CIFAR10 and CIFAR100 often diverge (Fig. 4: LRR with IMP mask vs IMP, Fig. 7), however the explanation for this divergence is not well understood. Although the authors list potential reasons, there is no experiment to support or reject the hypothesis. Evaluation on more datasets and more architectures should help to provide a stronger evidence for the claims and the relevance of the obtained theoretical findings.
* What is the meaning of the light blue line in Fig. 11?
* How does Fig. 12 support the claim made in the figure caption "LRR enables considerably more sign flips than IMP and thus improving mask identifcation and parameter optimization"?

---

> ### Author Response · Authors · 2023-11-15
> **Response to Reviewer yLRG**
>
> We sincerely thank the reviewer for their time and constructive feedback.
>
> **Response to weaknesses:**
>
> 1. We further confirm our findings on the larger and more complex datasets: Tiny ImageNet (see Figure 2d) and ImageNet (see Figure 15 in the appendix).
> 2. So far, LRR has only been proposed as a variant of IMP, which is the most common pruning baseline for its simplicity and high accuracy. It was therefore restricted to magnitude as a pruning criterion. Nevertheless, we have transferred the same principle to random pruning (see Figure 2. b,c and Figure 13 in the appendix), for which our insights hold as well. To extend the analysis, we have further added experiments for iterative Synflow and iterative SNIP in Fig. 16. The superiority of LRR can be generalized to all these different criteria. Also, our additional experiments with a VGG16 architecture on CIFAR10 in Figure 17 in the appendix are in line with these results.
> 3. In Figure 8 and 9 in the appendix, we provide a comparison on the effect of different learning rate schedules and highlight the benefits of using warmup epochs at the beginning of every training cycle to improve performance of both IMP and LRR.
>
> Minor: We have updated the draft with the corrected typo and added the references.
>
> **Response to questions:**
>
> 1. Experimental insights:
> - The main insight we draw from Figure 4 is that LRR is able to improve parameter optimization for different sparse masks, like the mask found by IMP (IMP mask). This finding actually holds for multiple datasets and learning settings. LRR with the IMP mask achieves a significantly higher performance than IMP on CIFAR10. If we modify LRR with BN parameter rewinding (red line) this holds also true on CIFAR100. Furthermore, note that we can draw similar conclusions for Tiny ImageNet (Fig. 2d), ImageNet (Fig. 15), different pruning criteria (Fig. 16), and VGG networks (Fig. 17).
>
> - Fig 7(c, d) additionally indicates that LRR depends significantly on the learned sign information, as sign perturbations can greatly hurt its performance. Yet, it is robust to magnitude rewinding (IMP reset amplitude in Fig 7). LRR signs and masks seem to contain sufficient information to achieve the high LRR performance. This conclusion is supported by the results on all studied datasets.
>
> 2. The blue line in Fig 11 (Fig 12 in the updated draft) refers to experiments, where we fix several aspects that have been learned by LRR. Concretely, we train an architecture with LRR mask and initial parameters that have the same signs as the learned LRR parameters. The only part that is still learned in the iterative scheme is the parameter magnitude, which starts from the initial IMP magnitude. The fact that this blue line achieves a similar performance as LRR suggests that the high LRR performance can primarily be attributed to the learned signs and the mask.
>
> 3. Figure 12 (13 in the updated draft) plots the difference between the total number of signs flipped from the initialization. It shows that LRR flips many more signs in comparison with IMP. We have removed the conclusion in the caption, as not only this figure but the collection of different experiments provide evidence for the stated hypothesis.

---

> > ### Comment · Reviewer_yLRG · 2023-11-21
> >
> > I appreciate the authors' comprehensive rebuttal response, which addressed my main concerns. The inclusion of additional experiments on various datasets, network architectures, and pruning criteria has significantly strengthened the paper. As a result, I have decided to revise my score upwards.

---

> > > ### Author Response · Authors · 2023-11-22
> > >
> > > We thank the reviewer for the constructive feedback that improved our paper and the resulting positive assessment of our work.

---

### Official Review · Reviewer_z4v8 · 2023-11-01

**Soundness:** 3 good
**Presentation:** 4 excellent
**Contribution:** 3 good
**Rating:** 6
**Confidence:** 4

**Summary:**

This paper delves into the effectiveness of learning rate rewinding (LRR) both from a theoretical and practical perspective, with a specific focus on the concept of weight sign flipping. In the context of a single-layer neural network, the authors offer theoretical evidence that LRR can reliably rectify initially problematic weight sign patterns by inheriting the sign information from a well-trained overparameterized model. When applied to more complex neural networks, empirical findings validate that the observed changes in weight sign configurations align with the theoretical insights. Furthermore, the authors dissect the impact of LRR into two distinct components: sparse network optimization and the performance of the generated mask. Through rigorous experiments, they provide empirical support for LRR's exceptional performance in both of these aspects.

**Strengths:**

- This paper offers a clear rationale for the effectiveness of LRR by examining it through the perspective of weight sign flipping, which is a pioneering work in exploring sign flipping within the context of sparse neural networks. The authors provide theoretical evidence demonstrating that LRR gains an advantage from flexible mask sign configurations, in contrast to IMP, a finding substantiated by empirical experiments at a toy-level. This research has the potential to serve as a source of inspiration for the advancement of more efficient sparse training algorithms that can leverage the power of mask sign configurations.
- The paper presents carefully designed ablation studies investigating two distinctive roles of LRR in (i) sparse neural network optimization and (ii) discovering a good sparse mask.
- The paper is effectively structured and exhibits clear and concise writing
- The paper covers a fair amount of relevant previous studies.

**Weaknesses:**

- While the authors argue LRR finds a better mask than WR in Figure 3, I wonder if a longer training epochs within each IMP cycle would help WR to find a superior mask. In other words, are both WR and LRR fully converged? If that’s the case, does the mask configuration stay constant after convergence? Further, if the optimal mask can be attained only at the end of the training epochs, it could pose challenges in efforts to reduce the computational cost associated with IMP (both WR & LRR).
- Concerning the flexible LRR mask analysis (see Figure 4), there appear to be some questionable findings. For instance, the "LRR with IMP mask (blue)" does not appear to show significant improvement over WR (orange) except for the case of Cifar-10 with a moderate level of sparsity. Moreover, it is unclear regarding the implication of "LRR w/ BN rewind (yellow)" in Figure 4 in the context of "flexible LRR training."
- In Figures 3 and 6, there are only two sets of experimental results available for analysis, Cifar-10 and Cifar-100. The authors argue that Cifar-100 results may not fully meet expectations due to its higher complexity. However, it remains an open question whether the same trend would hold for different network architectures, such as VGG networks.

**Questions:**

- Is there any further results on ImageNet or any large-scale datasets?
- In Figure 3, is the presented LRR results with or without BN rewinding?

---

> ### Author Response · Authors · 2023-11-15
> **Response to Reviewer z4v8**
>
> We sincerely thank the reviewer for their time and constructive feedback.
>
> **Response to weaknesses:**
>
> 1. Mask identification of IMP and LRR.
> - Both IMP and LRR are trained until convergence in each prune-train iteration (All details on the procedure are provided in Appendix 3 (Table 1)). The mask is identified by pruning the smallest parameters at the end of each training level once the model has converged. Hence it stays constant for each training iteration between pruning levels. The mask can also be identified after fewer training epochs optimizing a trade-off between model accuracy and computational costs (see e.g. [1,2]). However, we chose to identify the mask after convergence, because our work is focused on insights into the mechanisms that make iterative pruning algorithms succeed.
> - These insights might inspire computationally less expensive variants of LRR and IMP in future. For instance, we find the fact encouraging that LRR identifies good sign configurations relatively early in training.
>
> 2. Insights from Experiments are significant across learning tasks.
> - The main insight we draw from Figure 4 is that LRR is able to improve parameter optimization for different sparse masks, like the mask found by IMP (IMP mask). This finding actually holds for multiple datasets and learning settings. As correctly noted, LRR with the IMP mask achieves a significantly higher performance than IMP on CIFAR10. If we modify LRR with BN parameter rewinding and use it to train a network with the IMP mask (red line) this holds also true on CIFAR100. Note that the yellow line is not designed to showcase the flexibility of LRR in optimizing different masks, the red line is. In summary, the comparison of the red and the orange lines support our hypothesis.
>
> - Furthermore, note that we can draw similar conclusions for ImageNet (Fig. 15), different pruning criteria (Fig. 16), and VGG networks (Fig. 17).
>
> 3. We have added experiments for VGG16 in Fig. 17 in the appendix. The same general trends transfer also to this setting.
>
> **Response to questions:**
> 1. We have added experiments for ImageNet in Fig. 15 in the appendix. The results are consistent with our previous findings on CIFAR10, CIFAR100, and Tiny Imagenet.
> 2. Figure 3 is without BN rewinding in LRR. However, note that IMP always rewinds BN parameters.
>
> [1] You, Haoran, et al. "Drawing Early-Bird Tickets: Toward More Efficient Training of Deep Networks." International Conference on Learning Representations. 2019.
>
> [2] Paul, Mansheej, et al. "Unmasking the Lottery Ticket Hypothesis: What's Encoded in a Winning Ticket's Mask?." International Conference on Learning Representations. 2022.

---

> > ### Comment · Reviewer_z4v8 · 2023-11-22
> >
> > Thank you for the additional experiments!
> >
> > - I have one more lingering question regarding the early identification of good sign configuration. In [You et al., 2019], the early bird tickets emerge approximately less than 20% of the total training epochs. How early did the good sign configuration emerge?
> >
> > As the authors addressed my concerns, I will keep my rating and vote for acceptance.

---

> > > ### Author Response · Authors · 2023-11-22
> > > **Response to Question**
> > >
> > > We thank the reviewer for their feedback and interesting additional question for discussion.
> > >
> > > **On early sign identification:**
> > >
> > > We have verified that most of the signs for both LRR and IMP are identified by the $20$-th training epoch for each pruning level on CIFAR10. Concretely, on average, 75% of the signs are identified by the $20$-th epoch for both LRR and IMP. This finding is in line with You et al [1] and implies that early-bird ticket identification could result in good sign configurations and point towards a way to reduce the computational costs associated with LRR and IMP.
> > >
> > > [1] You, Haoran, et al. "Drawing Early-Bird Tickets: Toward More Efficient Training of Deep Networks." International Conference on Learning Representations. 2019.

---

### Official Review · Reviewer_6o5w · 2023-11-02

**Soundness:** 3 good
**Presentation:** 4 excellent
**Contribution:** 3 good
**Rating:** 6
**Confidence:** 3

**Summary:**

This study compares two key techniques in deep neural networks: Learning Rate Rewinding (LRR) and Iterative Magnitude Pruning (IMP). A clear and detailed analysis of both methods highlights the benefits of LRR, particularly its early parameter sign switching and better optimization of various network structures.
Through practical testing on different models and datasets, the authors present LRR's advantages, emphasizing it as a more versatile method for neural network optimization. This research serves as practical groundwork for further exploration of improvements to sparse training algorithms.
Interestingly, the authors examine the impact of sign perturbations. The experimental evidence in the paper shows that at lower sparsity levels, the impact of sign perturbations is small, but it has a significant effect on performance in the complex CIFAR100 dataset. This evidence further aligns with the authors' point - LRR is not only better at identifying masks but also optimizes various mask scenarios.

**Strengths:**

This paper presents two new advantages of Learning Rate Rewinding (LRR) and validates their existence, offering fresh insights not covered in previous LRR research. This work is an essential foundation for understanding and improving network pruning algorithms, particularly the LRR method. One notable contribution is the discovery of parameter sign switching, a key characteristic of LRR. This not only reveals a unique facet of LRR but also offers new perspectives for understanding and designing more effective algorithms.
The article presents enough experiments as evidence. Firstly, it uses a single hidden neuron model for learning tasks, showing that LRR has more success cases than IMP because LRR can avoid problematic initial sign configurations.
The paper further validates LRR's advantage over IMP through a series of representative tasks and networks. Experimental results show that LRR performs well in standard benchmark tests such as CIFAR10, CIFAR100, and Tiny ImageNet, regardless of its combination with ResNet18 or ResNet50. These results strongly support the superiority of LRR over IMP in deep neural network training.

**Weaknesses:**

This paper impressively combines clear presentation with strong experimental results, and I have yet to identify any significant shortcomings. But I have some questions about hyperparameters :
1. Considering that hyperparameter tuning is generally a problem-specific task, do you believe this sensitivity might hinder the practical application of LRR? Could you provide some advice or guidelines for hyperparameter selection or tuning when using LRR and discuss how the learning rate schedule influences the performance of this algorithm?
2. Your paper also indicates that LRR can benefit from the overparametrization of neural networks. Can you elaborate on how this overparametrization impacts the functionality of LRR? Is it possible to have too much overparametrization, which could negatively impact the performance of LRR?

**Questions:**

See above weakness.

---

> ### Author Response · Authors · 2023-11-15
> **Response to Reviewer 6o5w**
>
> We sincerely thank the reviewer for their time and constructive feedback.
>
> 1. Both IMP and LRR appear to be similarly sensitive to hyperparameters. In general, generalization on CIFAR10 is less sensitive to different learning rate schedules. However, for the larger CIFAR100 dataset, generalization is more susceptible to a poorly tuned learning rate schedule for iterative pruning. Both IMP and LRR have been used with a step or cosine learning rate schedule (Renda et al. [1]). We find that adding a few warmup epochs (usually 10) at each pruning-training iteration (with linearly increasing LR during warmup) improves the overall performance marginally on CIFAR10 and significantly on CIFAR100, for both LRR and IMP (see Figures 8 and 9 in the appendix). Hence, we propose to use warmup epochs for all experiments.
>
> 2. LRR benefits from overparameterization.
>   - Our theoretical insights on the single neuron model highlight that LRR can leverage overparameterization in early prune-train iterations to enable sign switches of initial parameters and find stable sign configurations which ensure that LRR succeeds more often than IMP. While an increasing input dimension does not harm LRR, at some point its positive impact saturates and further overparameterization is not required anymore (see Figure 18 in the appendix).
>    - Also experimentally, ResNet architectures tend to benefit from an increased width [2]. Yet, this benefit saturates at a certain point (so that the performance does not improve significantly by further increasing the width). This benefit also translates to pruning the corresponding architectures.
>  - We are not aware of a case, in which width overparameterization harms performance. However, we could imagine that it could cause issues with trainability and machine precision in extreme cases, but this would be a different topic of investigation.
>
> [1] Alex Renda, Jonathan Frankle, and Michael Carbin. Comparing rewinding and fine-tuning in neural network pruning. In International Conference on Learning Representations, 2020.
>
> [2] Golubeva, Anna, Guy Gur-Ari, and Behnam Neyshabur. "Are wider nets better given the same number of parameters?." International Conference on Learning Representations. 2020.

---

### Author Response · Authors · 2023-11-15
**Summary of Rebuttal**

We thank all reviewers for their insightful comments and the great effort that they put into the reviews. To provide an overview of our response to the suggestions, we summarize our updates of the manuscript and highlight our main contributions in the following.

**Updates to the manuscript:**
1. ImageNet: We provide results on the lmageNet dataset that are in line with our findings for other datasets. LRR’s ability to enable early sign flips also improves mask learning and parameter optimization in this large scale setting. See Figure 15 in the appendix.
2. Pruning criteria: To demonstrate that we provide insights into a general learning mechanism, we have conducted additional experiments with other common pruning criteria, specifically, Iterative Synflow and Iterative SNIP. Learning rate rewinding also enables more effective sign flips than weight rewinding in this context and leads to better performance.
3. We show that our results also transfer to other architectures like a VGG16 (see Figure 17 in the appendix).
4. For completeness, Figure 7 (d) includes now also results for CIFAR100 with IMP init + LRR mask + LRR sign (blue curve).
5. We show the effect of increasing input overparameterization for a single hidden neuron network (see Figure 18 in the appendix). Increasing overparameterization aids the optimization of LRR by allowing to correct initial bad signs.

**Main contributions:**
1. We have obtained novel insights on the importance of parameter sign switches during sparse training that are induced by initial overparameterization.
2. Based on theoretical and empirical evidence, we have thus identified a mechanism that makes Learning Rate Rewinding (LRR) more effective than Iterative Magnitude Pruning (IMP). This provides relevant insights into the standard baseline for contemporary neural network pruning algorithms.
3. Disentangling parameter optimization and mask identification, we show that LRR improves both these aspects by quickly switching initial signs. Our results are consistent across datasets (CIFAR10, CIFAR100, Tiny ImageNet, and ImageNet), different architectures (ResNet and VGG), and different pruning criteria (magnitude, random, SNIP and Synflow).

As our insights could lay the foundation to developing more competitive sparse training methods, we believe that our work is of high relevance for the ICLR community and would welcome any further questions.

---

### Meta-Review · Area_Chair_23F3 · 2023-12-11

**Metareview:**

The paper investigates two popular methods for finding sparse lottery tickets in deep networks: iterative magnitude pruning (IMP) and learning rate rewinding (LRR). It compares the two methods empirically and theoretically on a toy 1-hidden layer (single hidden neuron) network, and presents a thorough understanding of why LRR performs better than IMP. It also conducts ablations to disentangle the two aspects of structure (mask) learning and parameter optimization. One of the interesting findings is that good performance of LRR is mainly due to learned mask and parameter signs, while the parameter magnitudes are not that relevant. Authors have added new experiments on additional architectures and datasets to address some concerns from the reviewers. Overall this is a strong paper that makes a meaningful contribution to the understanding of an important problem of finding sparse lottery tickets and training sparse networks.

**Justification For Why Not Higher Score:**

All reviewers acknowledge that the paper makes an important contribution to the understanding of the sparse lottery tickets but none of the reviewers are enthusiastic enough to support a full oral presentation.

**Justification For Why Not Lower Score:**

AC believes this is a strong paper that adds significantly to the understanding of the sparse lottery tickets problem, and particularly why LRR works much better than IMP in training sparse subnetworks. This is a relevant problem that will be of interest to the broader deep learning community (as also pointed out by reviewer yLRG).

---

### Decision · Program_Chairs · 2024-01-16

Accept (spotlight)